

# Prognostic value of neutrophil-lymphocyte ratio in gastroenteropancreatic neuroendocrine neoplasm: a systematic review and meta-analysis

Yajie Wang[1], Bei Wen[2], Yuxin Zhang[3], Kangdi Dong[2], Shubo Tian[2] and Leping Li[2]

[1] Gastrointestinal Surgery, Peking University First Hospital, Beijing, China
[2] Gastrointestinal Surgery, Shandong Provincial Hospital Affiliated to Shandong First Medical University, Jinan, China
[3] Gastroenterology, Peking University Third Hospital, Beijing, China

## ABSTRACT

**Purpose**. A high neutrophil-to-lymphocyte ratio (NLR) might be connected with an unfavorable tumor prognosis. We sought to conduct a meta-analysis of published studies exploring the prognostic value of NLR in patients with gastroenteropancreatic neuroendocrine neoplasm (GEP-NEN).

**Methods**. We have referred to the PRISMA 2020 for the Abstracts checklist and have registered our review at the International Prospective Register of Systematic Reviews (registration number CRD42020187679). The PubMed, Embase, and Web of Science databases were screened using words like 'neutrophil to lymphocyte ratio', 'neuroendocrine tumors', and others up to July 2024. In our study, we evaluated the significance of NLR on overall survival (OS), recurrence-free survival (RFS), and progression-free survival (PFS) of patients with GEP-NEN. Subgroup analysis were conducted to identify the origins of heterogeneity and examine the impact of factor grouping.

**Results**. We gathered 18 cohorts with 2,995 cases. All included studies were high quality, with Newcastle Ottawa Scale (NOS) scores ranging from 6 to 8. The pooled analysis revealed that a higher NLR related to worse OS (hazard ratio (HR): 4.59, 95% confidence interval (CI) [3.35–6.29], $p < 0.00001$) and poor RFS (HR: 4.05, 95% CI [2.78–5.90], $p < 0.00001$) in patients with GEP-NEN. Subgroup analysis of race, tumor sites, and therapy showed good predictive significance, however, NLR is not effective in predicting the overall survival time of non-operative patients.

**Conclusion**. This meta-analysis showed that a high NLR predicted poor OS, RFS, and PFS in patients with GEP-NEN and can be used as a promising predictor.

Corresponding author
Yajie Wang,
pkudocyoung@outlook.com

## INTRODUCTION

Neuroendocrine neoplasms (NEN) are a group of heterogeneous neoplasms that originated from peptidergic neurons and neuroendocrine cells. Gastroenteropancreatic neuroendocrine neoplasm (GEP-NEN) accounts for most of this type of tumor (*Modlin et al., 2008*). According to the population-based study using nationally representative data from the Surveillance, Epidemiology and End Results (SEER) program, the incidence and prevalence of GEP-NEN are steadily rising, particularly in the small intestine, followed by rectum, appendix, colon, and stomach (*Patel, Barbieri & Gibson, 2019*; *Dasari et al., 2017*). Due to the high heterogeneity of this type of neoplasm, practical prognostic evaluation is critically needed.

In recent years, researchers found that inflammatory response played a decisive role in different stages of tumor development, including initiating, promoting, malignant transformation, invasion, and metastasis (*Grivennikov, Greten & Karin, 2010*). On the one hand, tumors change their microenvironment by secreting a variety of cytokines, and chemokines to weaken the systemic immune response and promote tumorigenesis and progression; on the other hand, systemic and local tissues are also infiltrated by immune cells and cytokine secretion to alter the tumor microenvironment and kill tumor cells (*Hinshaw & Shevde, 2019*). The neutrophil-to-lymphocyte ratio (NLR), as the ratio of neutrophils to lymphocytes in the peripheral blood, not only reflects the systemic tumor-associated inflammatory response but may also reflect bone marrow *versus* lymph, innate *versus* adaptive immunity, chronic inflammation *versus* acute immune rejection, tumor and antitumor immune equilibrium (*Park & Lopes, 2019*).

Here, we sought to conduct a systematic review and meta-analysis of published studies exploring the relationship between NLR with the prognosis in patients with GEP-NEN. We expected that NLR might be an available prognostic factor that could be used in clinical practice. Portions of this text were previously published as part of a preprint (https://doi.org/10.21203/rs.3.rs-34559/v1).

## MATERIALS AND METHODS

### Literature search

We adopted the Preferred Reporting Items for Systematic Review and Meta-Analysis (PRISMA) statements (*Moher et al., 2009*), registered at the International Prospective Register of Systematic Reviews (registration number CRD42020187679). This registration record has undergone automated eligibility checks and is published exactly as submitted. PubMed, Embase, and Web of Science electronic databases were searched for publications until July 2024. Combined text and MeSH terms were used for searching and the detailed search strategies are described below:

**Embase:** ('neuroendocrine tumor'/exp OR 'neuroendocrine tumors'/exp OR 'tumor, neuroendocrine' OR 'tumors, neuroendocrine') AND ('neutrophil to lymphocyte ratio'/exp OR 'neutrophil-to-lymphocyte ratio'/exp OR 'neutrophil/lymphocyte ratio'/exp OR 'NLR'). **PubMed:** ((Neuroendocrine Tumors) OR (Neuroendocrine Tumor) OR (Tumor, Neuroendocrine) OR (Tumors, Neuroendocrine)) AND (''neutrophil lymphocyte

ratio" or "neutrophil to lymphocyte ratio" or "neutrophil-to-lymphocyte ratio" or "neutrophil/lymphocyte ratio" or "NLR"). **Web of Science:** ((AB= (Neuroendocrine Tumors) OR AB= (Neuroendocrine Tumor) OR AB= (Tumor, Neuroendocrine) OR AB= (Tumors, Neuroendocrine)) AND ((AB= (neutrophil-lymphocyte ratio)) OR AB=(NLR)) OR AB= (neutrophil-to-lymphocyte ratio).

We include all possible eligible studies and consider them for review before screening, regardless of their primary outcome. We collected articles that were only written in English. Ethical approval was unnecessary because the meta-analysis is based on published clinical trial data.

## Inclusion and exclusion criteria

Two investigators (YW and BW) independently assessed all the potentially qualified articles at different times. If there are different views on inclusion or exclusion, we resolve them through discussion or consultation with other authors (YZ and KD). Titles and abstracts were reviewed first to determine whether studies were related to the theme. Then, full articles were judged according to the inclusion and exclusion criteria. If studies satisfied the inclusion criteria, they were used for detailed analysis and data extraction.

The inclusion criteria for qualified studies were as follows: (1) Patients included in the study were pathologically confirmed GEP-NEN in any stages and classification; (2) Assessment of OS, RFS, and PFS as outcomes; (3) The text reported the data of adjusted hazard ratio (HR) with 95% confidence intervals (CI), or any other type of survival information that can be calculated into HR with 95% CI; (4) The text reported an exact cut-off value of the NLR; (5) Articles published in English.

The exclusion principles were as shown below: (1) Lack of distinction of GEP-NENs from other NENs; (2) Duplicated studies or overlapping cohorts from the same centers; (3) Conference abstracts, reviews, case reports, mete-analysis, letters, animal studies, or laboratory studies; (4) Lack of necessary data.

## Data extraction and quality assessment

We extracted the following related information from each included study: name of the first author, published year of the article, study area, maximum tumor size, the period of the research, gender and sex ratio, age, study design type, intervention methods, the cut-off value of the NLR, HR and 95% CI of overall survival (OS), recurrence-free survival (RFS), and progression-free survival (PFS). In the meta-analysis, we preferred the adjusted HR values to maintain data consistency.

The OS time was figured out from the date of treatment initiation to the date of death. If the patient were still alive at the last follow-up, the endpoint is the date of the last follow-up. The RFS time was calculated as the number of months from the date of treatment to the date of confirmation of disease recurrence or the date the endpoint was realized. The PFS is the time from treatment to the first occurrence of disease progression or death from any cause. All the time is measured in a month for the unit. The NLR was calculated based on pretreatment laboratory data using the white blood cell differential counts by dividing the neutrophil count by the lymphocyte count. OS, RFS, and PFS outcomes were expressed as hazard ratio (HR) (and 95% CI) for patients with high NLR *versus* patients with low NLR.

We performed the quality assessment for the included studies according to the Newcastle–Ottawa scale (NOS). The full score is nine points, and studies with over five points were regarded as high-quality studies.

## Statistical analysis

We used the software of Review Manager (RevMan 5.3) for all statistical analyses. The pooled HR with its 95% CI was utilized to quantitatively assess the prognostic function of the NLR for GEP-NEN patients with the method described by Parmar (*Parmar, Torri & Stewart, 1998*). Cochrane Q and $I^2$ tests were used to evaluate the heterogeneity among the included studies. *P*-value $< 0.10$ for the Q test or $I^2 > 50\%$ indicates apparent heterogeneity, and we will choose the random-effects model. Otherwise, if *P*-value $\geq 0.10$ for the Q test or $I^2 \leq 50\%$, the fixed-effects model will be taken. We performed a subgroup analysis based on the geographical location of the studies, and tumor site if necessary. The stability of the results was confirmed by sensitivity analysis. $p < 0.01$ was considered statistically significant.

# RESULTS

## Characteristics of the included studies

The initial search contained 530 studies. After the removal of duplicates, 136 studies were excluded. Of the remaining 394 studies, 288 were eliminated by reading results (titles and abstracts) for apparent irrelevance. 46 full-text articles were downloaded to assess their eligibility, of which 33 were excluded because they were not English language ($n = 6$), ineffective NLR or OS data collected ($n = 17$), without adjusted HR data ($n = 5$), review ($n = 4$), and not GEP-NEN ($n = 1$). Ultimately, 13 studies (*Abdelmalak et al., 2021*; *Arima et al., 2017*; *Cao et al., 2017*; *Gaitanidis et al., 2018*; *Harimoto et al., 2019*; *Luo et al., 2017*; *Miura et al., 2021*; *Panni et al., 2019*; *Pozza et al., 2019*; *Yang et al., 2024*; *Yucel et al., 2013*; *Zhang et al., 2019*; *Zhou et al., 2017*) published between 2013 and 2024 were included in this meta-analysis. The sample sizes ranged from 48 to 620. The total sample size of our meta-analysis was 2,040 (Fig. 1).

The characteristics of the included studies are summarized in Table 1; Among them, 12 studies (*Abdelmalak et al., 2021*; *Arima et al., 2017*; *Cao et al., 2017*; *Harimoto et al., 2019*; *Luo et al., 2017*; *Miura et al., 2021*; *Panni et al., 2019*; *Pozza et al., 2019*; *Yang et al., 2024*; *Yucel et al., 2013*; *Zhang et al., 2019*; *Zhou et al., 2017*; *Arima et al., 2017*; *Harimoto et al., 2019*; *Zhang et al., 2019*) are retrospective cohort studies, one (*Gaitanidis et al., 2018*) is a prospective study. In terms of the research area, five studies (*Luo et al., 2017*; *Yang et al., 2024*; *Zhou et al., 2017*; *Zhang et al., 2019*; *Cao et al., 2017*) were conducted in China, two (*Gaitanidis et al., 2018*; *Panni et al., 2019*) in the USA, three (*Arima et al., 2017*; *Harimoto et al., 2019*; *Miura et al., 2021*) in Japan, one (*Abdelmalak et al., 2021*) in the UK, one (*Pozza et al., 2019*) in Italy, and one (*Yucel et al., 2013*) in Turkey. Regarding tumor site, eight studies concentrated on pancreatic neuroendocrine neoplasm (P-NEN), one study enrolled patients with gastric neuroendocrine neoplasm (G-NEN), three studies one study enrolled patients with gastrointestinal and pancreatic neuroendocrine neoplasms (GEP–NEN), one study only included enteric neuroendocrine neoplasms (E-NEN). A total

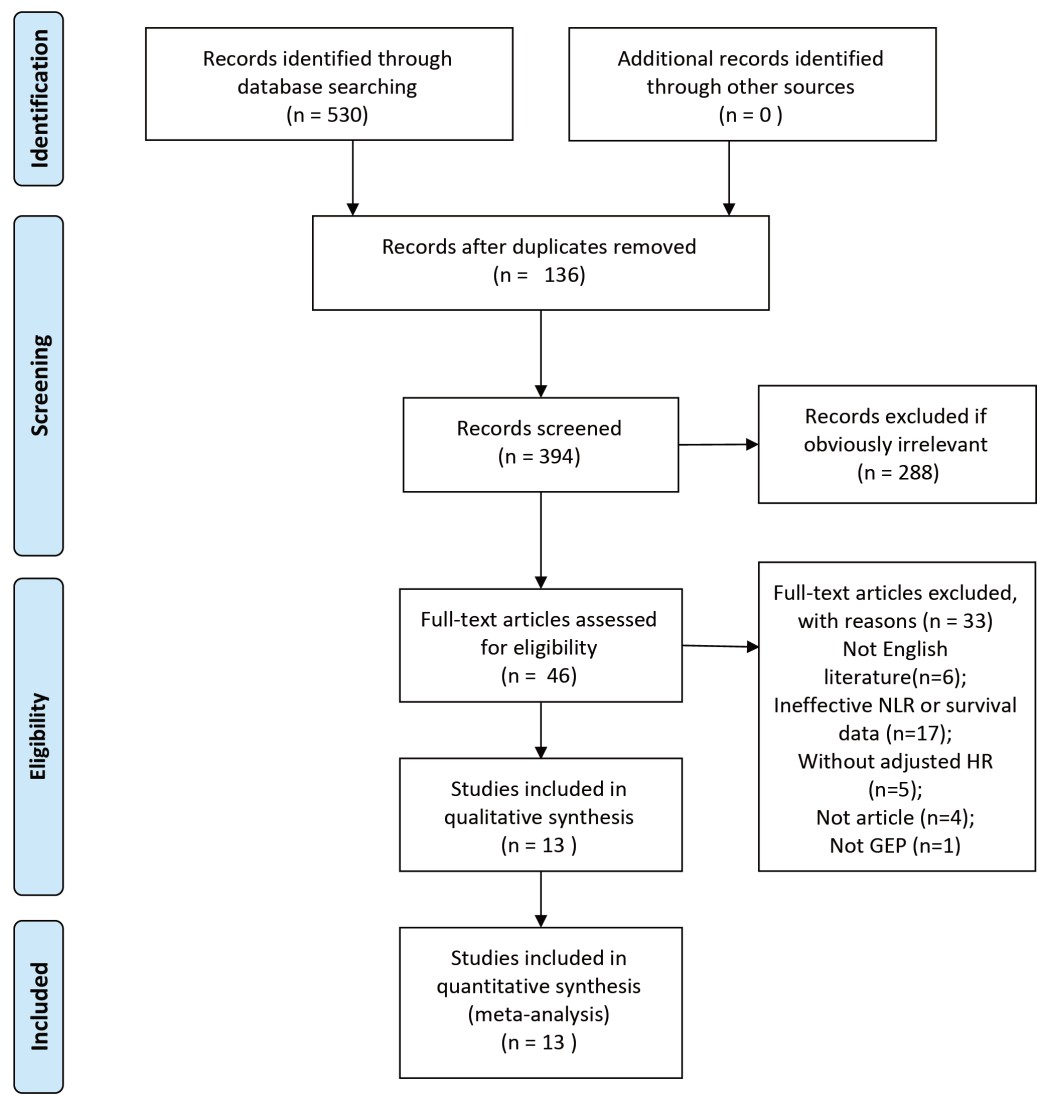

**Figure 1** PRISMA flowchart of the study screening process.

of 13 studies selected 1.79, 1.9, 2.2, 2.28, 2.3, 2.4, 2.4, 2.4, 2.62, 2.63, 3.41, 5 and 5 as the cut-off value of the NLR, respectively.

## Relationship between NLR with OS, RFS, and PFS

Nine studies embraced HR and 95% CI for OS. As shown in Fig. 2, a higher NLR was significantly associated with worse OS in the overall population with a pooled HR of 2.75 (95% CI [2.16–3.85], $p < 0.00001$). The heterogeneity analysis among the studies showed an $I^2$ value of 0% ($P = 0.58$), which indicated a low heterogeneity. Eight studies included risk ratio (HR) and 95% CI for RFS. Overall, higher NLR results in poorer RFS regardless of the effect model. In the random effects model, the pooled HR was 3.70 (95% CI [2.13–6.43], $p < 0.00001$) with high heterogeneity ($I^2$ value of 67%, $P = 0.006$). Only one study reported HR and 95% CI data for PFS.

**Table 1 The characteristics of the included studies.**

| Study | Area | Sample size | Gender (M/F) | Age (years) | Study period | Study design | Site | Therapy | Cut-off value of NLR | Survival analysis | HR (95% CI) | Confounding factors |
|---|---|---|---|---|---|---|---|---|---|---|---|---|
| Abdelmalak et al., 2021 | England | 77 | 36/41 | Median 63.1 | 2016–2020 | Retrospective | GEP | Chemotherapy | 5 | OS | 1.2 (0.4–3.5) | PS, Ki-67 index, mGPS, GI-NEC scores |
| Arima et al., 2017 | Japan | 58 | 27/31 | Median 58 | 2001–2015 | Retrospective | P | Surgery | 2.4 | RFS | 6.01 (1.84–19.64) | Tumor size |
| Cao et al., 2017 | China | 142 | 103/39 | <70, 109 cases and >70, 33 case | 2006–2015 | Retrospective | G | Surgery | 2.2 | OS | 2.61 (1.39–4.9) | Tumor size, Postoperative complication, Depth of invasion, Lymph node ratio, Ki-67 index |
| | | | | | | | | | | RFS | 2.97 (1.60–5.53) | Tumor size, Postoperative complication, Depth of invasion, Lymph node ratio, Ki-67 index |
| Gaitanidis et al., 2018 | America | 97 | 47/50 | Median 49.89 | NR | prospective | P | Surgery | 2.3 | RFS | 2.53 (1.05–6.10) | NR |
| Harimoto et al., 2019 | Japan | 55 | 23/32 | Median 61.08 | 2008–2017 | Retrospective | P | Surgery | 3.41 | RFS | 31.75 (1.93–522.33) | Synchronous hepatic resection, NET G2 or G3 vs NET G1 |
| Luo et al., 2017 | China | 165 | 38/51 | >50y, 69 cases and ≤50, 96 cases | 2006–2015 | Retrospective | P | Surgery and others | 2.4 | OS | 3.60 (1.33–9.74) | TNM stage, Grade, Symptom |
| Miura et al., 2021 | Japan | 120 | 49/71 | Median 60 | 2001–2018 | Retrospective | P | Surgery | 2.62 | RFS | 3.49 (1.05–11.60) | Tumor size, Clinical stage, 2017 WHO classification, Venous invasion |
| Panni et al., 2019 | United States | 620 | 321/299 | Median 57 | 2000–2016 | Retrospective | P | Sur, chemotherapy | 1.79 | RFS | 1.79 (1.20–2.67) | Tumor size, Adjuvant chemotherapy, Adjuvant radiation, Tumor stage |
| Pozza et al., 2019 | Italy | 48 | 26/22 | Median 67 | 2005–2016 | Retrospective | E | Surgery | 2.63 | OS | 4.71 (1.18–18.80) | NR |
| Yang et al., 2024 | China | 174 | 51.61 | 82/92 | 2009–2021 | Retrospective | P | NR | 2.28 | PFS | 1.69 (0.70–4.11) | Age, Subtype, Tumor size, Grade, LMR, PLR, CA19-9 |
| Yucel et al., 2013 | Turkey | 52 | 22/30 | ≥65y, 20 cases and '65y, 32 cases | 2006–2012 | Retrospective | G | Surgery | 5 | OS | 4.34 (1.20–15.70) | Surgical treatment, Grade |
| Zhang et al., 2019 | China | 260 | 100/56 | Mean 58 | 2000-2010 | Retrospective | G | Surgery | 2.4 | OS | 2.35 (1.24–4.42) | Age, Ki-67 index, Mitoses, Serum CEA/CA19-9, Distant metastasis |
| | | | | | | | | | | RFS | 8.00 (3.98–16.07) | Age, Ki-67 index, Mitoses, Serum CEA/CA19-9, Distant metastasis |
| Zhou et al., 2017 | China | 174 | 82/92 | Median 53 | 2008–2018 | Retrospective | P | Surgery | 1.9 | OS | 4.47 (1.53–13.06) | PLR, Grade, AJCC stage, LVSI Tumor size, AKT, Radical resection, Perineural invasion, Function, Symptomatic diagnosis |

**Notes.**

M/F, male-to-female; G, gastric neuroendocrine tumor; E, enteric neuroendocrine tumor; P, pancreatic neuroendocrine tumor; NR, Not reported; OS, overall survival; RFS, recurrence-free survival; PFS, progression-free survival.

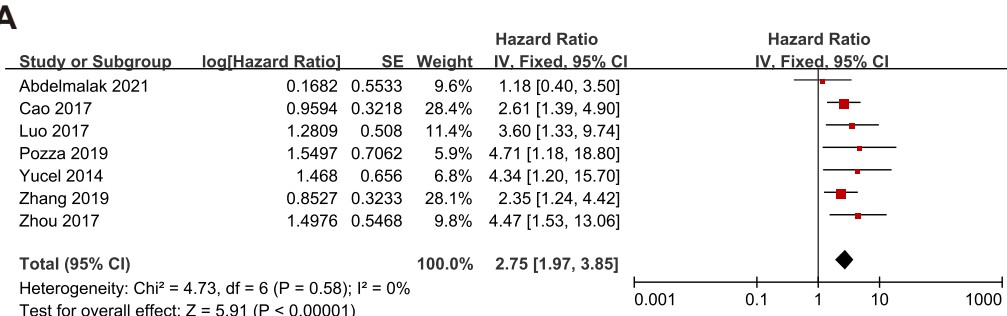

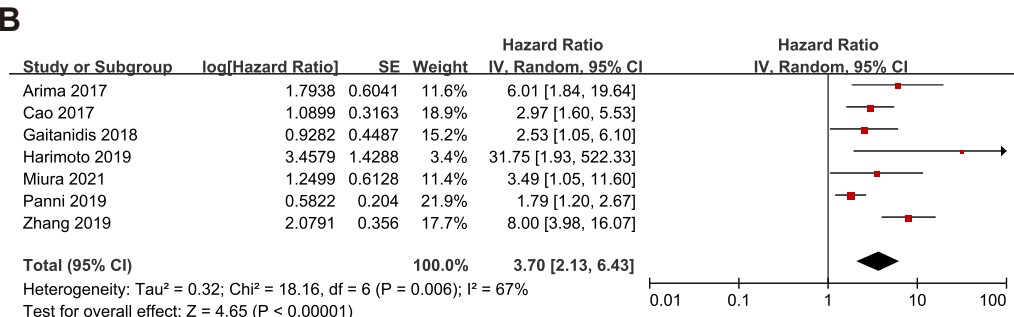

**Figure 2  Meta-analysis forest plot of relationships between NLR with (A)overall survival (B) recurrence-free survival.** Note: *Abdelmalak et al., 2021*; *Cao et al., 2017*; *Luo et al., 2017*; *Pozza et al., 2019*; *Yucel et al., 2013*; *Zhou et al., 2017*; *Arima et al., 2017*; *Gaitanidis et al., 2018*; *Harimoto et al., 2019*; *Miura et al., 2021*; *Panni et al., 2019*; *Zhang et al., 2019*.

## Subgroup analysis of the relationship between NLR and OS, RFS and PFS based on race

Since this meta-analysis was mainly based on Asian studies, we conducted OS subgroup analysis between Asians and Caucasians (Fig. 3). The subgroup analyses showed that high NLR was associated with a poor OS for patients in Asians (HR = 2.82, 95% CI [1.92–4.13], $p < 0.00001$) with a low heterogeneity ($I^2$ value of 0%, $P = 0.72$) and Caucasians (HR = 2.54, 95% CI [1.25–5.16], $p = 0.01$) with a low heterogeneity ($I^2$ value of 40%, $P = 0.19$).

The subgroup analysis demonstrated that an elevated NLR indicated a poor RFS in Asians (HR = 3.13, 95% CI [1.45–6.78], $p = 0.004$) with heterogeneity ($I^2$ value of 59%, $P = 0.06$) and Caucasians (HR=4.47, 95% CI [2.24–8.90], $p < 0.00001$) with heterogeneity ($I^2$ value of 55%, $P = 0.11$).

## Subgroup analysis of the relationship between NLR and OS, RFS and PFS in p-NEN and g-NEN

Since gastric neuroendocrine tumors (g-NEN) and pancreatic neuroendocrine tumors (p-NEN) accounted for the majority of the included literature, we conducted a subgroup analysis specifically for the prognostic analysis of NLR with g-NEN and p-NEN. The subgroup analysis pooled results showed that higher NLR was associated with worse OS (HR = 3.98, 95% CI [1.92–8.25], $p = 0.0002$), and RFS (HR = 2.27, 95% CI [1.63–3.16],

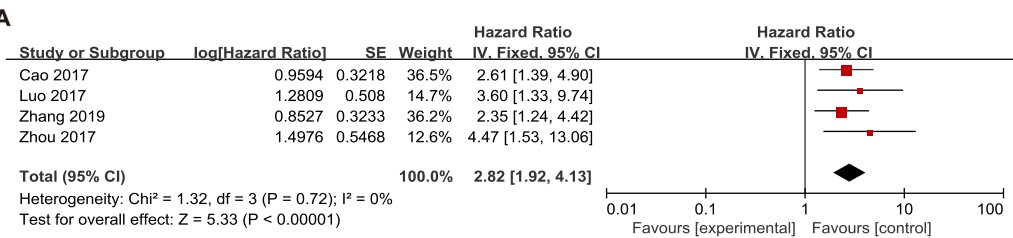

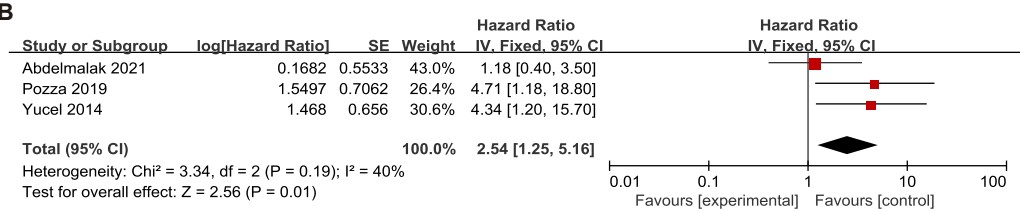

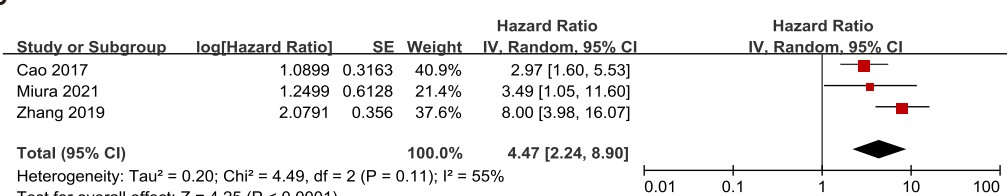

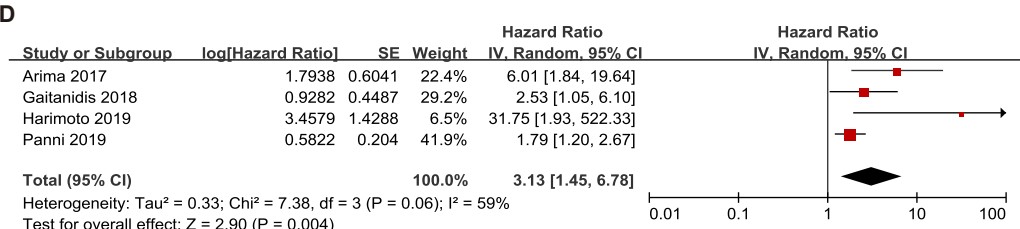

**Figure 3** Meta-analysis forest plot of the relationship of NLR with (A) overall survival (B) recurrence-free survival in Asians (C) overall survival (D) recurrence-free survival in Caucasians. Note: *Abdelmalak et al., 2021*; *Cao et al., 2017*; *Luo et al., 2017*; *Pozza et al., 2019*; *Yucel et al., 2013*; *Zhou et al., 2017*; *Arima et al., 2017*; *Gaitanidis et al., 2018*; *Harimoto et al., 2019*; *Miura et al., 2021*; *Panni et al., 2019*; *Zhang et al., 2019*.

$p < 0.00001$) of p-NEN (Figs. 4A–4B). The same results were found for NLR's prediction of g-NEN's OS (HR = 2.63, 95% CI [1.72–4.01], $p < 0.00001$) and RFS (HR=4.81, 95% CI [1.83–12.68], $p = 0.001$) (Figs. 4D–4E).

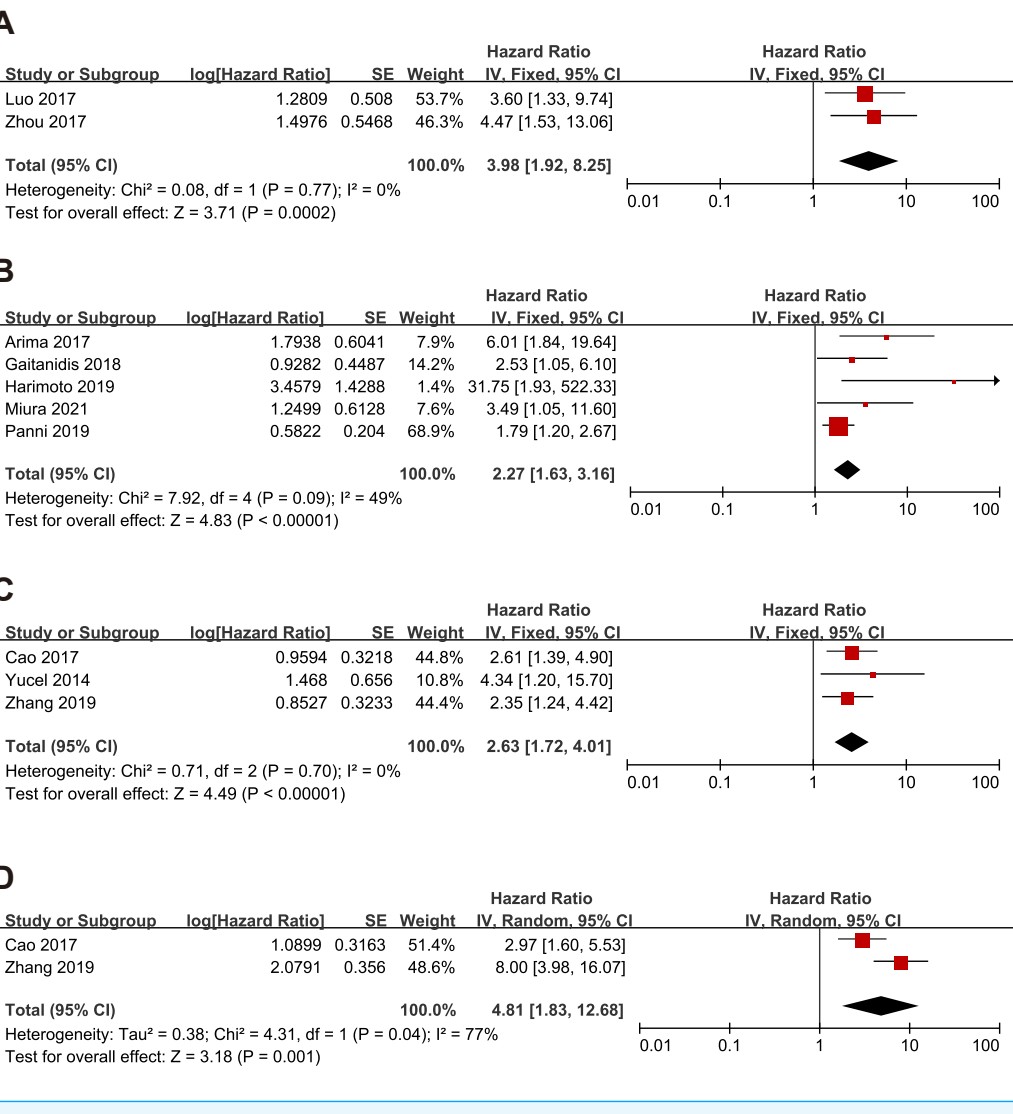

**Figure 4** Meta-analysis forest plot of the relationship of NLR with (A) overall survival, (B) recurrence-free survival in p-NEN, and (C) overall survival, (D) recurrence-free survival in g-NEN. Note: *Cao et al., 2017*; *Yucel et al., 2013*; *Arima et al., 2017*; *Gaitanidis et al., 2018*; *Harimoto et al., 2019*; *Miura et al., 2021*; *Panni et al., 2019*; *Zhang et al., 2019*; *Luo et al., 2017*; *Zhou et al., 2017*.

## Subgroup analysis of the relationship between NLR and OS, RFS and PFS in the surgery group and non-surgery group

We also examine the effect of NLR in surgical and non-surgical procedures. The subgroup analysis pooled results showed that higher NLR was associated with worse OS (HR = 2.93, 95% CI [2.01–4.28], $p < 0.00001$), RFS (HR = 4.29, 95% CI [2.97–6.18], $p < 0.00001$) in patients under surgery procedure (Figs. 5A–5B).

## Sensitivity analysis

We conducted a sensitivity analysis for the primary outcome of OS by individually excluding each study to assess the impact of each study on the overall results. We found that the

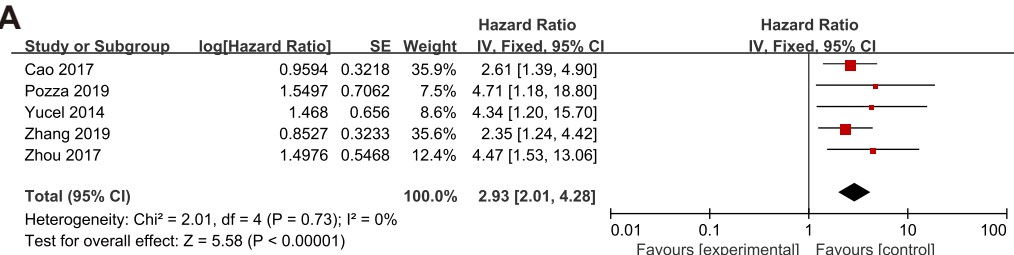

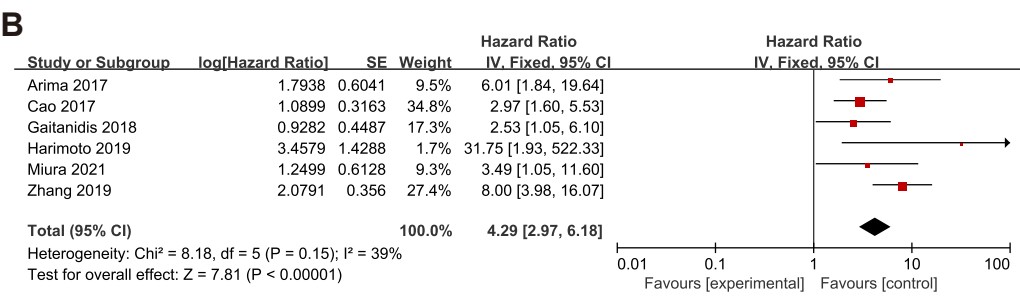

**Figure 5** Meta-analysis forest plot of the relationship of NLR with (A) overall survival and (B) recurrence-free survival in GEP-NEN patients under surgery procedure. Note: *Cao et al., 2017*; *Pozza et al., 2019*; *Yucel et al., 2013*; *Zhou et al., 2017*; *Arima et al., 2017*; *Gaitanidis et al., 2018*; *Harimoto et al., 2019*; *Miura et al., 2021*; *Zhang et al., 2019*.

removal of any single study did not significantly affect the final meta-analysis results or the heterogeneity results (see Fig. 6). Similarly, we removed each study individually to conduct a sensitivity analysis to assess each study's effect on the overall primary result for RFS. We found an apparent descending heterogeneity ($I^2$ value of 39% with $P = 0.15$) when we removed Panni's study, which might be the source of high heterogeneity (Fig. 7).

## Study quality

According to the NOS score, all the retrospective studies were high quality, ranging from 6 to 8 (Table 2).

## Publication bias

The shape of the funnel plots (Figs. 8A–8B) showed asymmetry and indicated significant publication bias in OS, RFS. We found that the funnel plot for publication bias in the OS outcome was symmetrical, indicating that there was essentially no publication bias. However, in the funnel plot for the RFS outcome, the study by Harimoto deviated from the symmetrical funnel, suggesting that this study had a significant publication bias.

## DISCUSSION

GEP-NEN is one of the most common types of neuroendocrine tumors. Some researchers have found that the prognosis is associated with factors like tumor classification, stage, and immunohistochemistry (*Massironi et al., 2014*; *Wang et al., 2012*). To enhance clinical

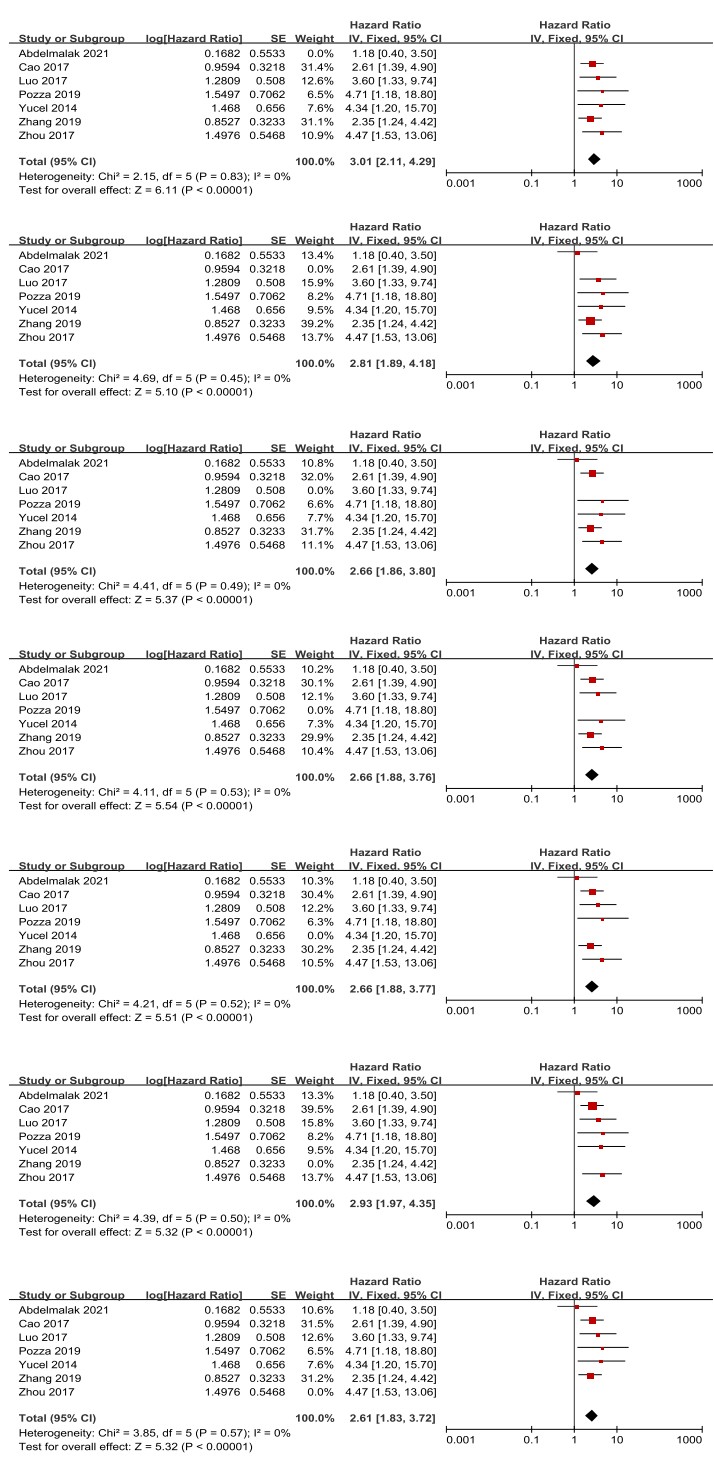

**Figure 6 Sensitivity analysis for the outcome of overall survival.** Note: *Abdelmalak et al., 2021*; *Cao et al., 2017*; *Luo et al., 2017*; *Pozza et al., 2019*; *Yucel et al., 2013*; *Zhou et al., 2017*; *Zhang et al., 2019*.

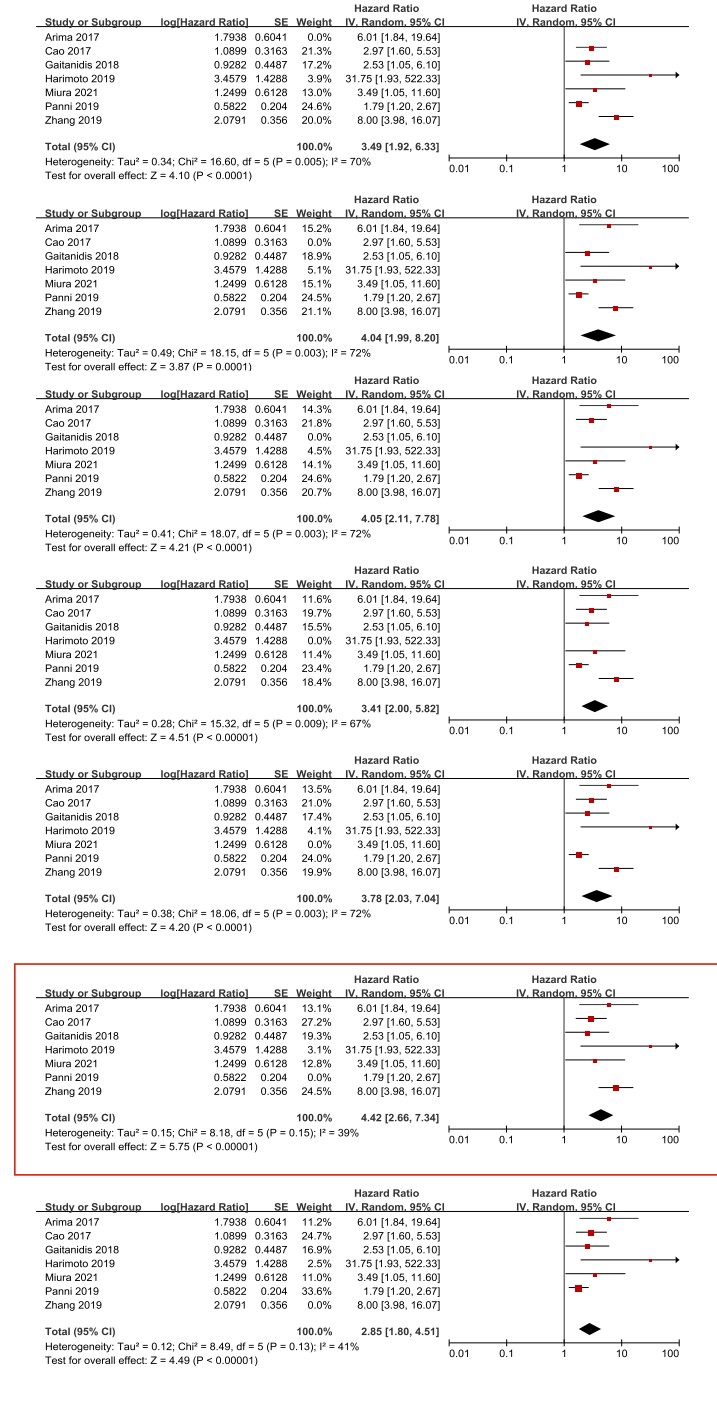

**Figure 7 Sensitivity analysis for the outcome of recurrence-free survival.** Note: *Cao et al., 2017*; *Arima et al., 2017*; *Gaitanidis et al., 2018*; *Harimoto et al., 2019*; *Miura et al., 2021*; *Panni et al., 2019*; *Zhang et al., 2019*.

**Table 2 Results of quality assessment using the NOS score for the included retrospective studies.**

| Study | Selection | | | Comparability | | Outcomes | | | Total quality scores |
|---|---|---|---|---|---|---|---|---|---|
| | Representativeness of the exposed cohort | Selection of the unexposed cohort | Ascertainment of exposure | Outcome of interest not present at start of study | Control for important factor or additional factor | Assessment of outcome | Follow-up long enough for outcomes to occur | Adequacy of follow-up of cohorts | |
| Abdelmalak et al. (2021) | 1 | 1 | 1 | 0 | 1 | 1 | 1 | 1 | 7 |
| Arima et al. (2017) | 1 | 1 | 1 | 0 | 1 | 1 | 1 | 0 | 6 |
| Cao et al. (2017) | 1 | 1 | 1 | 0 | 1 | 1 | 1 | 1 | 7 |
| Gaitanidis et al. (2018) | 1 | 1 | 1 | 0 | 1 | 1 | 1 | 0 | 6 |
| Harimoto et al. (2019) | 1 | 1 | 1 | 0 | 2 | 1 | 1 | 0 | 7 |
| Luo et al. (2017) | 1 | 1 | 1 | 0 | 1 | 1 | 1 | 1 | 7 |
| Miura et al. (2021) | 1 | 1 | 1 | 0 | 1 | 1 | 1 | 1 | 7 |
| Panni et al. (2019) | 1 | 1 | 1 | 0 | 1 | 1 | 1 | 1 | 7 |
| Pozza et al. (2019) | 1 | 1 | 1 | 0 | 1 | 1 | 1 | 0 | 6 |
| Yang et al. (2024) | 1 | 1 | 1 | 0 | 1 | 1 | 1 | 1 | 7 |
| Yucel et al. (2013) | 1 | 1 | 1 | 0 | 1 | 1 | 1 | 1 | 7 |
| Zhang et al. (2019) | 1 | 1 | 1 | 0 | 1 | 1 | 1 | 1 | 7 |
| Zhou et al. (2017) | 1 | 1 | 1 | 0 | 2 | 1 | 1 | 1 | 8 |

decision-making in oncology, there is an urgent need for a reliable and accessible prognostic biomarker.

In this meta-analysis, we synthesized the available literature to investigate the correlation between pretreatment neutrophil-to-lymphocyte ratio (NLR) and patient outcomes in GEP-NEN. After a rigorous selection process, our analysis encompassed 13 studies involving 2,040 patients, thereby evaluating the clinical relevance of NLR as a prognostic factor in GEP-NEN. Our findings suggest that an elevated pretreatment NLR is significantly correlated with poorer survival rates among GEP-NEN patients. This study underscores the potential of NLR as a valuable tool in predicting survival in GEP-NEN, highlighting its importance in the management of these tumors.

Several studies implied that an elevated NLR was associated with poor survival in several types of cancer, such as esophageal cancer (*Yodying et al., 2016*), breast cancer (*Ethier et al., 2017b*), colorectal cancer (*Haram et al., 2017*), prostate cancer (*Gu et al., 2016*) and ovarian cancer (*Ethier et al., 2017a*). The results of our pooled analysis focusing on GEP-NEN agreed with the results from these above-mentioned studies on other cancers. We can see that most of the included research is from China, which may cause selection bias; therefore, we performed a subgroup analysis of the studies based on race diffidence to explore whether race affected the results. We considered that a higher NLR is associated with reduced survival time in both Asians and Caucasians. We also performed subgroup analyses based on tumor sites (stomach and pancreas). Not surprisingly, NLR showed good predictive performance across all subgroup analyses. Therefore, NLR carries great potential to predict the prognoses in patients with GEP-NENs. Meanwhile, the subgroup analysis of patients with surgery methods, we found that NLR was good in predicting the survival time of surgical patients.

**A**

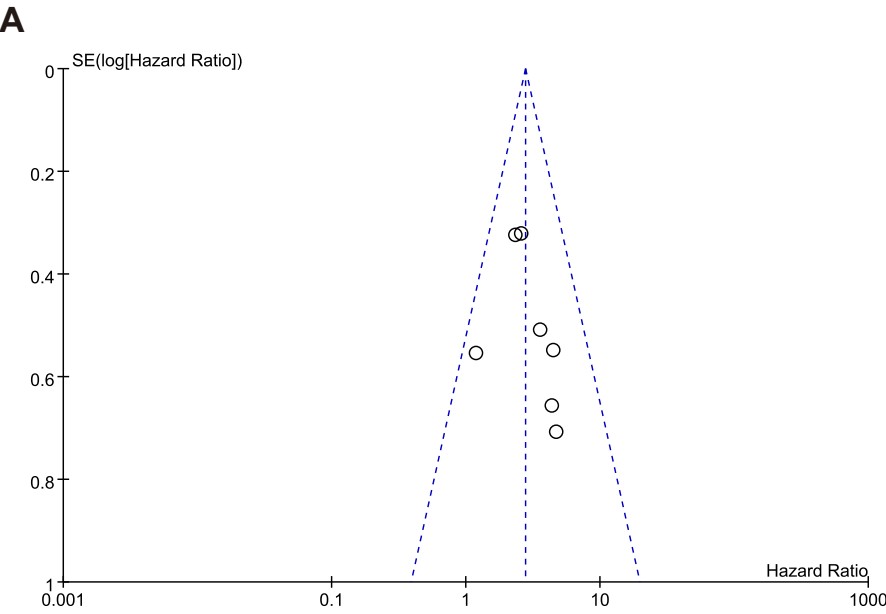

**B**

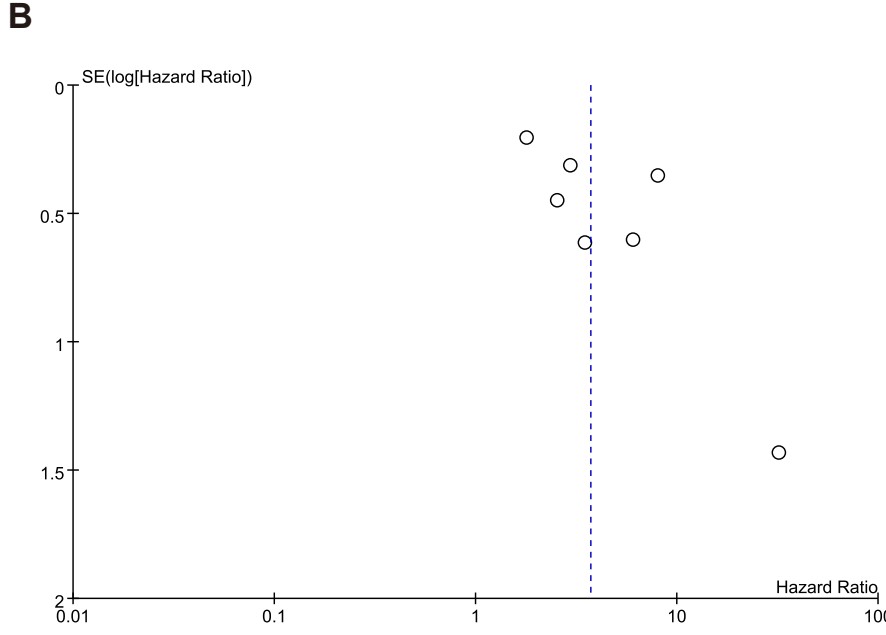

**Figure 8** **Funnel plot of publication bias test for (A) overall survival (B) recurrence-free survival.**

The relationship between chronic inflammation and cancer has been gradually known in recent years. A study showed that activated inflammatory cells in the gastric mucosa play a key role in the development of pernicious anemia (*Troilo et al., 2019*). However, the theories of the relationship between a high value of NLR and a worse prognosis in cancer patients have not been demonstrated. Neutrophils are the most abundant white blood cells in circulation and are the first responders to sites of infection and tissue damage.

Tumor-associated neutrophils (TANs) predict poor overall survival in many types of cancer (*Shaul & Fridlender, 2019*). Persistent inflammation promotes tumor growth, and the chemokines and cytokines, including CXCL8, CXCL5, and CXCL6 generated by tumor cells, and the surrounding microenvironment are involved in neutrophil recruitment (*Powell & Huttenlocher, 2016*). In a broad sense, lymphocytes have anti-tumor activity, so the reduction of lymphocytes is conducive to maintaining the tumor microenvironment and the growth of tumor cells. Tumor-infiltrating lymphocytes mainly include CD8+ cytotoxic T cells (CLT), CD4+ T cells (helper T cells), and a small number of regulatory T cells (Treg). CD8+ T cells (CTL) can directly recognize and kill tumor cells; CD4+ Cells (helper T cells) can secrete cytokines that assist CTL in killing tumor cells (*Hanahan & Coussens, 2012*). These are the possible mechanisms by which high NLR can predict tumor Invasive growth.

In the sensitivity analysis of NLR with RFS, after revaluation of *Panni et al.*'s (*2019*) that might be the source of heterogeneity. We found that in this large sample size cohort, there were also patients with late-stage tumors, which greatly affected the efficacy of NLR in predicting tumor prognosis. This indicated that the stage had a specific effect on the prognosis of GEP-NEN, which might be a confounding factor in NLR prognosis.

In our study, limitations existed with no doubt. First, 12 of 13 included researches were retrospective studies, and all 13 included studies were of small sizes. Second, most included patients underwent surgical treatment, some underwent chemotherapy and somatostatin treatment, which also resulted in a little heterogeneity. In addition, it is better to perform subgroup analysis by tumor grade, and stage to rule out their influence on the results. Finally, non-English language literature was not included, and there might be more valuable results that were not included.

## CONCLUSION

In conclusion, this meta-analysis showed that a high NLR was predictive of poor OS and RFS in patients with GEP-NEN. More high-quality prospective clinical trials are required to evaluate the feasibility of NLR in GEP-NEN.

**Abbreviations**

| | |
|---|---|
| **NLR** | neutrophil-to-lymphocyte ratio |
| **GEP-NEN** | gastroenteropancreatic neuroendocrine neoplasm |
| **OS** | overall survival |
| **RFS** | recurrence-free survival |
| **PFS** | progression-free survival |
| **HR** | hazard ratio |
| **CI** | confidence interval |
| **SEER** | Surveillance, Epidemiology and End Results |
| **PRISMA** | Preferred Reporting Items for Systematic Review and Meta-Analysis |
| **WBC** | white blood cell |
| **NOS** | Newcastle–Ottawa scale |
| **TANs** | tumor-associated neutrophils |

### Funding

The authors received no funding for this work.

### Competing Interests

The authors declare there are no competing interests.

### Author Contributions

- Yajie Wang conceived and designed the experiments, performed the experiments, analyzed the data, prepared figures and/or tables, authored or reviewed drafts of the article, and approved the final draft.
- Bei Wen performed the experiments, authored or reviewed drafts of the article, and approved the final draft.
- Yuxin Zhang analyzed the data, authored or reviewed drafts of the article, and approved the final draft.
- Kangdi Dong analyzed the data, prepared figures and/or tables, and approved the final draft.
- Shubo Tian analyzed the data, authored or reviewed drafts of the article, and approved the final draft.
- Leping Li conceived and designed the experiments, authored or reviewed drafts of the article, and approved the final draft.

### Data Availability

This is a systematic review/meta-analysis.

### Supplemental Information

Supplemental information for this article can be found online at http://dx.doi.org/10.7717/peerj.19186#supplemental-information.

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
