# Peer review of "Prognostic value of neutrophil-lymphocyte ratio in gastroenteropancreatic neuroendocrine neoplasm: a systematic review and meta-analysis"

_PeerJ, doi:10.7717/peerj.19186_

## Round 0.1 · original submission · Major Revisions

The authors are requested to carefully revise the manuscript and answer the questions raised by the reviewers.

Reviewer 1 ·

Basic reporting

The manuscript is overall well balanced; professional English language proofing would furthermore increase the quality;
- row 165 - discussion section - gynecologic cancers et al seems to be and error and should be corrected; - row 168 - the "showed" - is misspelled 
- tables - the width of the cells should be adjusted to make the text inside easily readable
- figures - title of the figure does not always correspond to the legends - figure 3 below the Forest plot should be OS not NLR;  Figure 4  and 7 should refer to PFS not OS; figure 5 - there is no control and experimental group, it should refer to PFS based on the reported results

Experimental design

The idea is original and relevant and the research question is well defined; the methods section could be improved to increase clarity:
- date of the systematic search is 4 years ago - the systematic search should be updated since relevant data might be missed;
- the search key should be found in the Annex however it is not available; it is important to include it in the manuscript;
- the authors claim to have included in their meta analysis only articles reporting hazard ratios; however survival data is sometimes reported in figures; have the authors considered extracting data from the figures or contacting the authors of these articles to obtain the raw data for calculating the HRs?

Validity of the findings

no additional comments;

Additional comments

The authors may consider the following additional comments:
in the abstract - there is no information about the statistical analysis and risk of bias assessment - based on PRISMA recommendations, they should be included;

PRISMA is a reporting guideline not a guideline for conducting a meta-analysis

Have the authors considered further subgroup analysis based on tumor location - ex for pancreatic NENs or tumor grade? it would surely be interesting to see the results for further evaluating the clinical scenario NLR is fit for prognostic assessment in GEP NENs;

Reviewer 2 ·

Basic reporting

Wang et al have conducted a systematic review and meta-analysis of literature on the prognostic significance of NLR for gastroenteropancreatic neuroendocrine tumors. Although this is a theme that deserves to be explored, the present article has some significant flaws. First of all, the article would benefit from substantial linguistic editing; in its current state even the title is grammatically incorrect. Secondly and most importantly, the authors have conducted a literature search up to May 2020 (line 9, line 47). This is more than 4 years old at the time of this review and should probably be updated to include the more recent literature before being considered for re-submission.

Experimental design

The stated aim of the study was to investigate the association between NLR and prognosis in patients with GEP-NEN, a valid and clinically relevant endpoint. However the GEP-NEN group is quite diverse. It includes any stages and classifications (line 59), all locations and various treatments (although most usual treatment was surgery (table 1). It would be interesting if the authors could examine the effect of NLR in more homogeneous subgroups (e.g. pancreas and siNET, non-metastatic and metastatic, surgical and non-surgical procedures).

The study has been registered in PROSPERO, PRISMA guidelines were followed, inclusion and exclusion criteria are described and studies were evaluated with the Ottawa scale. It was not clear which search strategy was used (in abstract: using words[….] and others, in line 48 it is referred to appendix but it doesn’t seem to be included in the downloaded files I have, apologize if I miss it somewhere). It is not clear why OS and RFS are defined in lines 68-71, were they calculated for the purpose of this study? Was the definition the same in all studies reviewed? As far as I can see it is only HRs/CIs and not RFS/OS that were analyzed? Also multivariable HRs were collected, and described in the results section (e.g. line 117) but it is unclear if they were used somewhere?

Validity of the findings

In lines 188-189 it seems as if the studies by Fan et al, Grenadier et al were excluded from analysis? this was not mentioned in the results section. The HRs presented are without these studies? What would the HRs be with these studies? This is an unexpected choice, as in table 1, these are the two by far largest studies, and both have a fine NOS score of 7.

The authors should state in the conclusion more clearly that RFS benefit is seen only in the Asian group.

---

## Round 0.2 · Major Revisions

The authors are requested to carefully further revise the manuscript and answer the questions raised by the reviewers.

Reviewer 1 ·

Basic reporting

During the revision the authors have implemented the suggestions that were made by updating the systematic search, performing additional clinically relevant subgroup analysis improved the readability of the figures and overall the quality of the manuscript. There is only one major aspect unapproached - English language proofreading - professional writing can always benefit from such a service and sometimes even authors who are native English speakers have difficulties in conveying their message in such a style; this manuscript now meets all other quality criteria for publication except the use of English language. I ensure the authors that this comment is exclusively aimed at improving the quality of the paper.

Experimental design

no comment

Validity of the findings

no comment

Reviewer 2 ·

Basic reporting

This updated version of the manuscript has considerably improved upon some of the issues noted in the previous version. The theme of the meta-analysis is relevant and interesting. Most importantly, the literature search has been updated adding several new studies published in the previous four years.
1.As noted in the previous version of this article, professional linguistic review would probably still be recommended (e.g. in the title neoplasms instead of neoplasm, in the abstract, “we have been referred to the PRISMA checklists” sounds syntactically wrong, hazard ratio instead of hazards ratio etc).

Experimental design

2.In the methods section, it is unclear what RFS and PFS are. I assume that RFS is taken for studies reporting recurrence (not progression as stated in text?) and PFS for studies reporting on medical therapies? Since only one study was prospective, PFS cannot be the time from randomization but rather from treatment start?
3. Additionally, the authors do not define their strategy for extracting HRs/Cis from the paper. Do they choose only reported HRs or are they also extracted for example from other sources (as in Parma et al that is referred in the text?) Are they using only unadjusted or only adjusted HRs or a mix of both? If both adjusted and unadjusted HRs are reported, is there a predefined strategy for selecting, as for example in https://doi.or g/10.1371/journal.pone.0263661 ? If unadjusted HRs are used, do the studies still get points in the NOS score for control for important cofounders?

Validity of the findings

4. When checking just the first three articles in the OS graph, Fig 2A, there seem to be some inaccuracies: In the Abdelmalak study the HR for OS in the whole cohort was not 1.6 as written in the graph (that was just for the NEC subgroup) but rather 1.2 (0.4-3.5) in the adjusted analysis. For the second study, by Abou-Jokh et al, adjusted HR is used in the meta-analysis (1/0.16= 6.25), instead of the lower unadjusted HR that is also reported. For the third study in the graph, by Cao et al, the higher unadjusted HR is used (2.94), instead of the lower adjusted HR which is also reported in the study (2.6). It is unclear what the authors’ criteria for selection are ?

5. In graph 2B, only four studies have an HR line whereas all other studies are weighted to 0%?

6. In lines 229-231, it is suggested that NLR is worse in predicting OS for late-stage and higher grade patients. Do the authors have some support in literature for this idea? It seems more likely that higher-grade tumors and tumors in later lines of treatment are more inflammatory and NLR would probably have better discrimination in just that group?

7. The sensitivity analysis as I understand excludes some of the larger studies. I think that was also the case in the previous version of the articles with the studies of Fan and Grenadier, it is unclear why these have been excluded from this version of the meta-analysis. Additionally in lines 246-255, the authors suggest that the reason for all heterogeneity is indeed the intrinsic differences between patients treated with surgery versus those treated with medical treatments (chemotherapy/SSA). If there is indeed such heterogeneity, the authors would probably best consider completely splitting these two groups and presenting two completely separate analysis?

---

## Round 0.3 · Minor Revisions

The authors are requested to carefully revise the manuscript and answer the questions raised by the reviewers.

Reviewer 1 ·

Basic reporting

The authors have revised the manuscript for the second time improving the use of professional English language.

Experimental design

this aspect was addressed during the firs revision round

Validity of the findings

this aspect was addressed during the firs revision round

Reviewer 3 ·

Basic reporting

Line 61, " the predictive role of NLR in GEP-NEN remains controversial". However, almost all the studies indicate that high NLR in GEP-NEN is associated with poor outcomes.

Experimental design

1. The retrieval strategy should be submitted as supplementary materials.
2. The number of references in Figure 1 is inconsistent, plz check them.

Validity of the findings

1. “In the meta-analysis, we preferred the adjusted HR values to maintain data consistency”, confounding factors and HR (95%CI) of each study should be provided in Table 1.
2. It could be “NR” (not reported), not UR in the study column of Table 1.
3. Funnel plot is not suitable for publication bias test anymore when the number of studies <10. Therefore, P-value of Begg’s test and Egger’s test could be provided if possible. Moreover, the authors described that a significant publication bias was observed in the current study, trim and fill method could be used to adjust the publication bias to re-evaluate the meta-analysis result.

Additional comments

None

---

## Round 0.4 · Minor Revisions

The authors are requested to carefully revise the manuscript and answer the final questions raised by the reviewer.

Reviewer 3 ·

Basic reporting

None

Experimental design

None

Validity of the findings

None

Additional comments

The authors did a good job of answering my questions. However, I still got some confusion. Firstly, why the authors conducted the sensitivity analysis for RFS only, not for both OS and RFS? The authors seem not to clarify it in the manuscript. Additionally, the sensitivity analysis was conducted by removing each study. I recommend the authors display all the results by removing one study each time. Thirdly, is it reasonable for NOS to replace the publication bias? Are there any publications that could support it? Are Begg's and Egger's tests not feasible in this study?

---

## Round 0.5 · accepted · Accept

After revisions, one reviewer agreed to publish the manuscript. There are two reviewers left with minor revision, and I think the author has responded adequately. I also reviewed the manuscript and found no obvious risks to publication. Therefore, I also approved the publication of this manuscript.